# FMRP regulates postnatal neuronal migration via MAP1B

**Salima Messaoudi[1], Ada Allam[1], Julie Stoufflet[1,2], Theo Paillard[1], Anaïs Le Ven[1,3], Coralie Fouquet[1], Mohamed Doulazmi[1], Alain Trembleau[1], Isabelle Caille[1,4]\***

[1]Sorbonne Université, CNRS UMR8246, Inserm U1130, Institut de Biologie Paris Seine (IBPS), Neuroscience Paris Seine (NPS), Paris, France; [2]Laboratory of Molecular Regulation of Neurogenesis, GIGA-Stem Cells and GIGA-Neurosciences, University of Liège, CHU Sart Tilman, Liège, Belgium; [3]Institut Curie, Paris, France; [4]Université de Paris, Paris, France

**\*For correspondence:**
isabelle.caille@u-paris.fr

**Competing interest:** The authors declare that no competing interests exist.

**Abstract** The fragile X syndrome (FXS) represents the most prevalent form of inherited intellectual disability and is the first monogenic cause of autism spectrum disorder. FXS results from the absence of the RNA-binding protein FMRP (fragile X messenger ribonucleoprotein). Neuronal migration is an essential step of brain development allowing displacement of neurons from their germinal niches to their final integration site. The precise role of FMRP in neuronal migration remains largely unexplored. Using live imaging of postnatal rostral migratory stream (RMS) neurons in *Fmr1*-null mice, we observed that the absence of FMRP leads to delayed neuronal migration and altered trajectory, associated with defects of centrosomal movement. RNA-interference-induced knockdown of *Fmr1* shows that these migratory defects are cell-autonomous. Notably, the primary *Fmrp* mRNA target implicated in these migratory defects is microtubule-associated protein 1B (MAP1B). Knocking down MAP1B expression effectively rescued most of the observed migratory defects. Finally, we elucidate the molecular mechanisms at play by demonstrating that the absence of FMRP induces defects in the cage of microtubules surrounding the nucleus of migrating neurons, which is rescued by MAP1B knockdown. Our findings reveal a novel neurodevelopmental role for FMRP in collaboration with MAP1B, jointly orchestrating neuronal migration by influencing the microtubular cytoskeleton.

## eLife assessment

This study addresses the role of FMRP in the migration of newborn neuroblasts in the postnatal brain. Through extensive and **convincing** analysis of living imaging videos, the authors showed that neurons with FMRP deletion migrate aberrantly and exhibit defects in nucleokinesis and centrokinesis. The study presents a **valuable** finding on the mechanism of neuroblast migration in the postnatal brain.

## Introduction

The fragile X syndrome (FXS) is the most common cause of inherited intellectual disability and a leading cause of autism spectrum disorder. FXS is due to the silencing of the gene *FMR1* and loss of the encoded protein, FMRP (fragile X messenger ribonucleoprotein) (*Davis and Broadie, 2017*). FMRP is an ubiquitous RNA-binding protein, with high level of expression in the central nervous system (*Gholizadeh et al., 2015*). It is a general regulator of RNA metabolism and especially of mRNA local translation in neurons (*Banerjee et al., 2018*). Its cognate mRNA targets are numerous and diverse, including mRNAs encoding proteins involved in neuronal plasticity like CamKIIα (calcium

calmodulin-dependent kinase II) and cytoskeletal proteins like microtubule-associated protein 1B (MAP1B) (*Ascano et al., 2012*; *Brown et al., 2001*; *Darnell et al., 2001*; *Maurin et al., 2018*). *Fmr1*-null mice are the murine model of FXS and have allowed characterization of numerous neurodevelopmental and plasticity defects consecutive to the absence of FMRP. We previously showed the essential role of FMRP in the differentiation and learning-induced structural plasticity of adult-generated olfactory bulb interneurons (*Scotto-Lomassese et al., 2011*; *Daroles et al., 2016*).

Neuronal migration is a crucial step for the establishment of neuronal circuitry, orchestrating the relocation of neurons from their birthplace to their final destination for differentiation. Migration defects lead to severe brain pathologies including lissencephaly and cortical heterotopia and may contribute to psychiatric disorders (*Romero et al., 2018*). Interestingly, migration in the human infant brain appears to be even more extended than anticipated from the rodent data (*Sanai et al., 2011*; *Paredes et al., 2016*). In addition, periventricular heterotopia has been documented in two individuals with FXS, implying a potential role for FMRP in migration (*Moro et al., 2006*). Importantly, in *Fmr1*-null mice, radially migrating embryonic glutamatergic cortical neurons display a defect in the multipolar to bipolar transition (*La Fata et al., 2014*), a critical change of polarity taking place before the initiation of the movement. The key *Fmrp* target mRNA involved in this process is N-cadherin, downregulated in the Fmr1 mutants and whose overexpression rescued the migratory phenotype. Additionally, both FMRP overexpression or knockdown (KD) lead to the misplacement of cortical glutamatergic neurons, also potentially underscoring its role in radial embryonic migration (*Wu et al., 2019*). However, to our knowledge, postnatal neuronal migration in the absence of FMRP has not been studied so far and the dynamics of mutated *Fmr1* neurons have yet to be comprehensively analyzed.

Here, using the postnatal rostral migratory stream (RMS) as a tangential migration model as in *Stoufflet et al., 2020*, we present novel insights into the migratory defects induced by the absence of FMRP. Migrating neurons in *Fmr1*-null mice exhibit a distinctive pattern of slowed-down and erratic migration, accompanied by alterations of centrosome movement. Notably, these defects are cell-autonomous, as evidenced by their recapitulation through RNA-interference-induced Fmr1 KD.

MAP1B is a microtubule-associated protein that plays a crucial role in the regulation of cytoskeletal dynamics within neurons. It is instrumental in brain development (*Villarroel-Campos and Gonzalez-Billault, 2014*) including neuronal migration (*Yang et al., 2012*; *González-Billault et al., 2005*; *Gonzalez-Billault et al., 2004*). We hypothesized its involvement as the *Fmrp* mRNA target in the observed defects. Indeed, elevated levels of MAP1B were detected in the mutated RMS, and the KD-induced downregulation of MAP1B successfully rescued most migration defects. Finally, we elucidate the molecular mechanisms underlying these migratory defects, by showing that the microtubular cage surrounding the nucleus of migrating neurons is disrupted in the absence of FMRP. Importantly, this abnormality is rescued by MAP1B KD, shedding light on a previously undiscovered role for the FMRP/MAP1B duo in regulating proper tangential migration at the microtubular level.

## Results

### FMRP is expressed in migrating neurons of the postnatal RMS

A massive number of neuroblasts are constantly produced in the ventricular/subventricular zone (V/SVZ) and migrate over a long distance along the RMS to the olfactory bulb (OB) (*Lim and Alvarez-Buylla, 2016*; *Figure 1A*). They display a cyclic saltatory mode of migration, in which the nucleus and centrosome move forward in a 'two-stroke' cycle (*Bellion et al., 2005*). The centrosome moves first within a swelling in the leading process, termed here centrokinesis (CK) followed by movement of the nucleus, referred to as nucleokinesis (NK) (*Figure 1B*). The neurons then pause and the cycle can reinitiate.

After an in vivo intraventricular electroporation of a GFP-expressing plasmid in neonate mice, a sub-population of GFP-positive neurons can be visualized in the RMS a few days later (*Figure 1C*).

FMRP is expressed in most neurons of the brain (*Gholizadeh et al., 2015*). Accordingly, immunostaining for FMRP reveals that FMRP is strongly expressed in the RMS, where most neurons appear labeled (*Figure 1D*). In individual GFP-positive neurons, FMRP labeling appears as a discrete and punctate staining visible mainly in the cytoplasm both at the rear of the neuron and in the leading process (*Figure 1E*). In order to more precisely localize FMRP at the subcellular level in migrating neurons, we performed culture of V/SVZ explants in Matrigel as described (*Wichterle et al., 1997*)

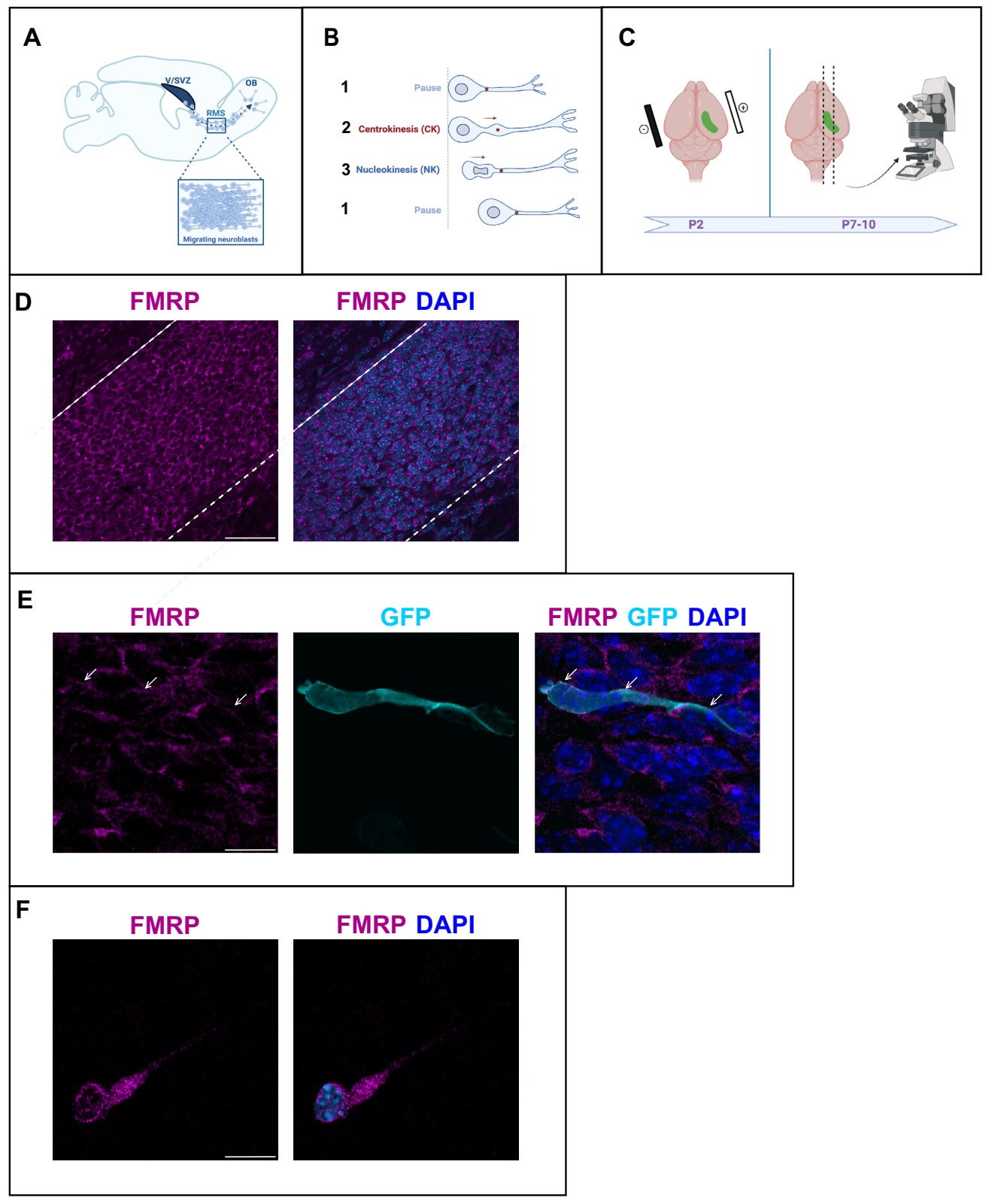

**Figure 1.** Fragile X messenger ribonucleoprotein (FMRP) is expressed in migrating neurons of the murine postnatal rostral migratory stream (RMS). (**A**) Scheme of a sagittal section of the postnatal RMS connecting the V/SVZ to the OB. V/SVZ, ventricular/subventricular zone; OB, olfactory bulb; RMS, rostral migratory stream. The inset shows the high density of homotypically migrating neurons in the RMS. (**B**) Representation of cyclic saltatory migration. (1) The neuron is in pause. (2) The leading process extends, and the centrosome moves within a swelling in the leading process. (3) The

*Figure 1 continued on next page*

Figure 1 continued

nucleus moves forward. CK, centrokinesis; NK, nucleokinesis. (C) Scheme of the experimental procedure. 2-day-old neonates are intraventricularly electroporated with a GFP-expressing plasmid to label a cohort of migrating neurons that can be subsequently visualized in fixed or acute sections of the RMS. (D) Immunohistochemistry of the RMS showing FMRP expression (magenta) along the stream. The RMS is delineated with dotted lines. Scale bar: 50 µm. (E) Immunohistochemistry of a GFP-positive RMS neuron (cyan) showing FMRP subcellular expression (magenta). The GFP-positive neuron displays a cytoplasmic expression of FMRP around the nucleus (indicated by white arrows). The surrounding GFP-negative neurons express FMRP as well, following the same pattern. Scale bar: 10 µm. (F) Immunostaining for FMRP of a neuroblast migrating away from a V/SVZ explant in Matrigel. The labeling appears granular in the cytoplasm around the nucleus, in the swelling and the leading process. Scale bar: 10 µm.

followed by FMRP immunostaining (*Figure 1F*). FMRP labeling appeared as a granular staining similarly visible around the nucleus, in the swelling and in the leading process.

## FMRP cell-autonomously regulates neuronal migration

To investigate the involvement of FMRP in RMS migration, we used the *Fmr1*-null mouse line (*Bakker et al., 1994*). Time-lapse imaging of GFP-positive neurons was performed in the control and *Fmr1*-null RMS (*Figure 2—video 1*, *Figure 2—video 2* from https://osf.io/eqhzx).

*Fmr1*-null neurons display a slowed-down migration, an increased pausing time, a more sinuous trajectory, and a defective directionality (*Figure 2A–D* from https://osf.io/eqhzx). Additionally, the NK is less frequent and the mean distance per NK is reduced (*Figure 2E and F* from https://osf.io/eqhzx).

Given the crucial role of the centrosome in neuronal migration (*Higginbotham and Gleeson, 2007*), we analyzed its dynamics by performing co-electroporation of GFP and centrin-RFP in *Fmr1*-null and control neonate mice in order to co-label migrating neurons and their centrosome (*Figure 3—video 1* from https://osf.io/eqhzx). The CK is slowed down and less frequent in *Fmr1*-null neurons, as compared to controls (*Figure 3A and B* from https://osf.io/eqhzx). A CK leading to a subsequent NK was defined as an efficient CK, as opposed to a CK not leading to an NK. CK efficiency is reduced in *Fmr1*-null neurons as compared to controls (*Figure 3C* from https://osf.io/eqhzx).

To determine whether the observed migration defects are cell-autonomous, we designed an interfering RNA coupled to GFP to cell-autonomously knock down *Fmr1* mRNA in RMS neurons, similar to *Scotto-Lomassese et al., 2011*. Its efficiency was assessed by FMRP immunostaining on electroporated SVZ explants in Matrigel. While all neurons electroporated by the control miRNeg-GFP were FMRP immunoreactive, this was reduced to 36% in miRFmr1-GFP electroporated neurons (Chi2 test, p-value = 0.01; miRNeg condition: N=3, n=15; miRFmr1 condition: N=3, n=14). The interfering miR*Fmr1*-GFP was thus co-electroporated with centrin-RFP to perform live imaging. Analysis of migration and centrosome dynamics (*Figure 2—figure supplements 1 and 2* from https://osf.io/eqhzx) showed that *Fmr1* KD is sufficient to mostly recapitulate the migratory phenotype described in *Fmr1*-null mutants, revealing that FMRP cell-autonomously regulates neuronal migration (Kruskal-Wallis followed by Dunn's post hoc analysis on the four genotypes). The only discernible difference lies in the directionality parameter, where defects are exacerbated in KD neurons compared to *Fmr1*-null mutants (Fisher's exact test p<0.001). This suggests that this defect might not be cell-autonomous in *Fmr1*-null mutants but rather a consequence of the mutated environment. This more pronounced directionality defect in the KD could be indicative of a lack of compensation in the acute KD context.

Together, these data demonstrate that FMRP is cell-autonomously necessary for the proper neuronal migration of RMS neurons.

## FMRP regulates neuronal migration through MAP1B

MAP1B is a neuron-specific microtubule-associated protein widely expressed in the developing CNS with crucial roles in diverse steps of neural development including neuronal migration (*Yang et al., 2012*; *González-Billault et al., 2005*; *Gonzalez-Billault et al., 2004*).

Immunostaining of the RMS revealed MAP1B expression, with most neurons appearing labeled (*Figure 4A*). MAP1B subcellular staining is consistent with labeling of microtubules both around the nucleus and in the leading process (*Figure 4A*). Immunostaining of individualized neurons migrating in Matrigel similarly showed a labeling around the nucleus and in the leading process, with occasional microtubule bundles (*Figure 4B*).

MAP1B is a well-described *Fmrp* mRNA target (*Zhang et al., 2001*; *Darnell et al., 2001*; *Brown et al., 2001*). Given its expression in RMS migrating neurons, it emerged as an interesting target for

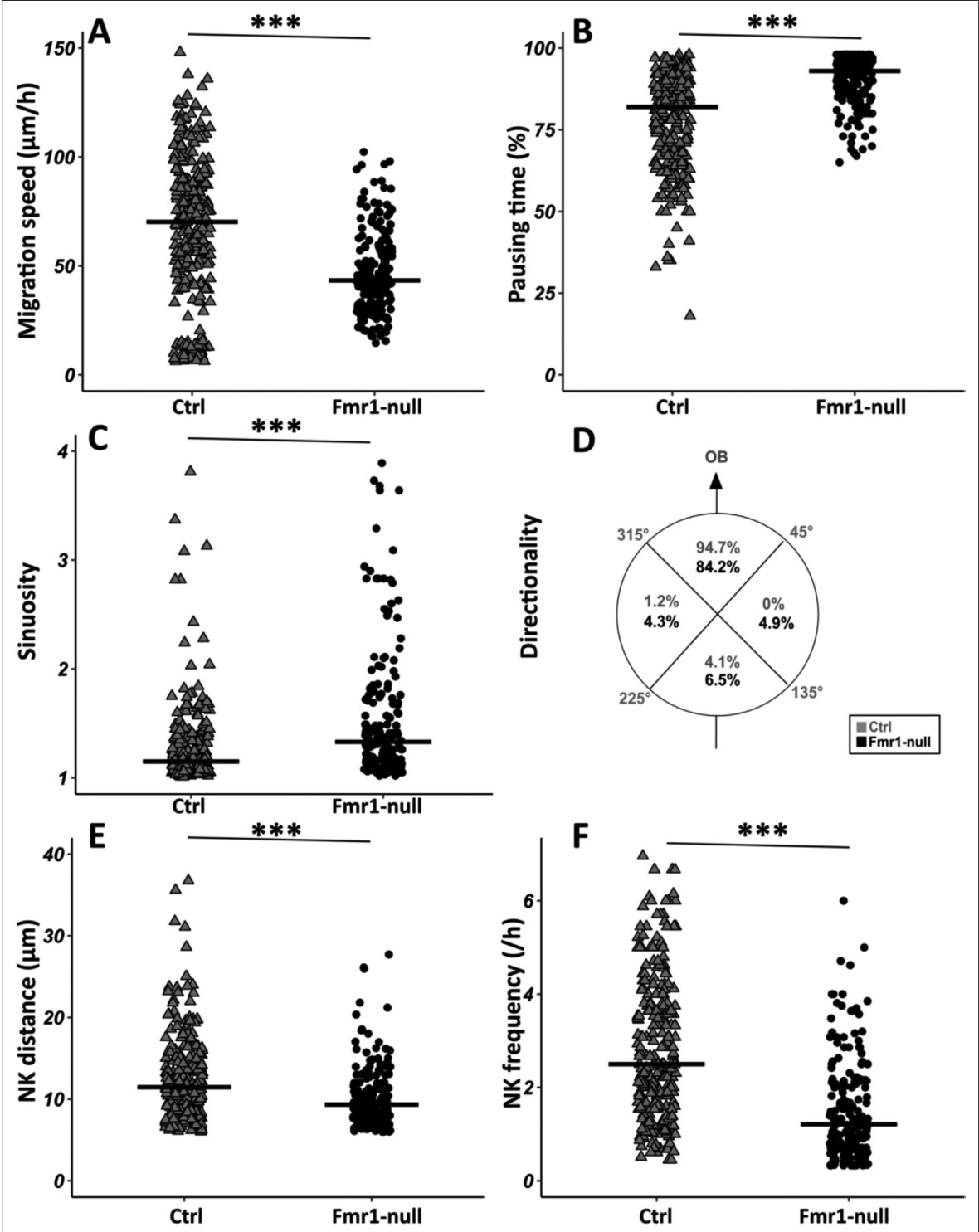

**Figure 2.** Migration defects in *Fmr1*-null neurons. (**A**) Migration speed of control (Ctrl) and *Fmr1*-null neurons. Ctrl: 70.62 (43.32) μm/hr; *Fmr1*-null: 43.34 (25.97) μm/hr (Kruskall-Wallis test: Chi2=91.92, p-value <0.001, df = 3; followed by Dunn's post hoc test). (**B**) Percentage of pausing time of control and *Fmr1*-null neurons. Ctrl: 82 (21.50); *Fmr1*-null: 93 (9.25) (Kruskall-Wallis test: Chi2=130.61, p-value <0.001, df = 3; followed by Dunn's post hoc test). (**C**) Sinuosity index of control and *Fmr1*-null neurons. Ctrl: 1.15 (0.26); *Fmr1*-null: 1.35 (0.66) (Kruskall-Wallis test: Chi2=65.19, p-value <0.001, df = 3; followed by Dunn's post hoc test). (**D**) Migration directionality radar represented in four spatial dials. Percentage of cells migrating in each spatial direction in control and *Fmr1*-null neurons, relatively to the vector going straight from the subventricular zone (SVZ) to the olfactory bulb (OB) (Fisher's exact test, p-value <0.001). (**E**) Nucleokinesis (NK) mean distance of control and *Fmr1*-null neurons. Ctrl: 11.46 (6.27) μm; *Fmr1*-null: 9.34 (4.16) μm (Kruskall-Wallis test: Chi2=53.45, p-value <0.001, df = 3; followed by Dunn's post hoc test). (**F**) NK frequency of control and *Fmr1*-null neurons. Ctrl:

*Figure 2 continued on next page*

*Figure 2 continued*

2.5 (2.23) NK/hr; *Fmr1*-null: 1.21 (1.45) NK/hr (Kruskall-Wallis test: Chi2=111.53, p-value <0.001, df = 3; followed by Dunn's post hoc test). The black line represents the median. Ctrl: N=3, n=275; *Fmr1*-null: N=3, n=184. Median (IQR). ***p-value <0.001. Source videos, tracking data (NK frequency, pausing time, sinuosity, speed, NK distance, directionality), and statistical analysis of migrating control and Fmr1-null neurons are available at https://osf.io/eqhzx.

The online version of this article includes the following video and figure supplement(s) for figure 2:

**Figure supplement 1.** Fmr1 knockdown (KD) recapitulates *Fmr1*-null neuronal migration defects.

**Figure supplement 2.** Fmr1 knockdown (KD) recapitulates *Fmr1*-null neurons centrokinesis (CK) defects.

**Figure 2—video 1.** Time-lapse imaging of GFP electroporated neurons in control rostral migratory stream (RMS). https://elifesciences.org/articles/88782/figures#fig2video1

**Figure 2—video 2.** Time-lapse imaging of GFP electroporated neurons in Fmr1-null rostral migratory stream (RMS). https://elifesciences.org/articles/88782/figures#fig2video2

further investigation. As FMRP is a repressor of *Map1b* mRNA translation (*Brown et al., 2001*; *Darnell et al., 2001*; *Lu et al., 2004*), the overall level of MAP1B typically appears increased in an *Fmr1*-null context (*Lu et al., 2004*; *Hou et al., 2006*). Accordingly, the quantification of three independent western blots showed that MAP1B expression is increased on average by 1.6× in the RMS of *Fmr1*-null mice compared to controls (*Figure 4—figure supplement 1* from https://osf.io/eqhzx).

To investigate whether the observed migratory phenotype in *Fmr1*-null neurons is influenced by the upregulation of MAP1B, we cell-autonomously knocked down *Map1b* in RMS neurons with an interfering RNA. Its efficiency was assessed by MAP1B immunostaining on electroporated SVZ explants in Matrigel, similar to the miRFmr1-GFP. While all neurons electroporated by the control miRNeg-GFP were MAP1B immunoreactive, this was reduced to 46% in miRMAP1B-GFP electroporated neurons (Chi2 test, p-value <0.001. miRNeg condition: N=4, n=103; miRMap1b condition: N=3, n=113). The miR*Map1b*-GFP plasmid was electroporated in *Fmr1*-null neonate mice and time-lapse imaging was conducted on acute sections of the RMS (*Figure 5—video 1* from https://osf.io/eqhzx). *Fmr1*-null neurons expressing miR*Map1b*-GFP exhibited a complete restoration of migration speed, pausing time, NK distance, and frequency, making them comparable to miRNeg-GFP control neurons (*Figure 5A, B, D, E* from https://osf.io/eqhzx). Notably, the sinuosity of *Fmr1*-null neurons expressing miR*Map1b*-GFP was not rescued (*Figure 5C* from https://osf.io/eqhzx), suggesting that this parameter is MAP1B-independent.

In conclusion, our results demonstrate that MAP1B is the primary *Fmrp* mRNA target responsible for regulating neuronal migration.

## The FMRP/MAP1B duo acts on the microtubular cage of RMS neurons

Considering the microtubule-associated functions of MAP1B, we investigated whether the migratory phenotype observed in Fmr1 mutants could be linked to a compromised microtubular cytoskeleton. To test this, we employed intraventricular electroporation of a plasmid expressing doublecortin (DCX) fused to RFP for labeling the microtubules of RMS neurons (*Koizumi et al., 2006*).

In control neurons co-electroporated with mirNeg-GFP, subsequent immunostaining revealed a well-defined cage of DCX-positive microtubule bundles smoothly encircling the nucleus in the vast majority of neurons, consistent with observations in other systems (*Figure 6A* from https://osf.io/eqhzx; *Rivas and Hatten, 1995*; *Shu et al., 2004*; *Xie et al., 2003*). Contrastingly, mirFmr1-GFP co-electroporation with DCX-RFP unveiled an aberrant microtubular cage in a majority of neurons (*Figure 6B* from https://osf.io/eqhzx). These disorganized neurons exhibited disruptions in the cage (46% of them) or sinuous cages (33%) and sometimes even bundles of microtubules detaching from the nucleus (21%). Prompted by these findings, we investigated whether MAP1B overexpression in Fmr1 mutants was responsible for this microtubular aberration. Co-electroporation of DCX-RFP with mirFmr1-GFP and mirMAP1B-GFP indeed demonstrated a rescue of the abnormal cage in these double mutants (*Figure 6C*, quantification in *Figure 6D* from https://osf.io/eqhzx).

In conclusion, our findings uncover the migratory phenotype of Fmr1 mutants, attributing it to an anomalous microtubular cage caused by MAP1B overexpression, elucidating a critical interplay between FMRP, MAP1B, and the microtubular cytoskeleton.

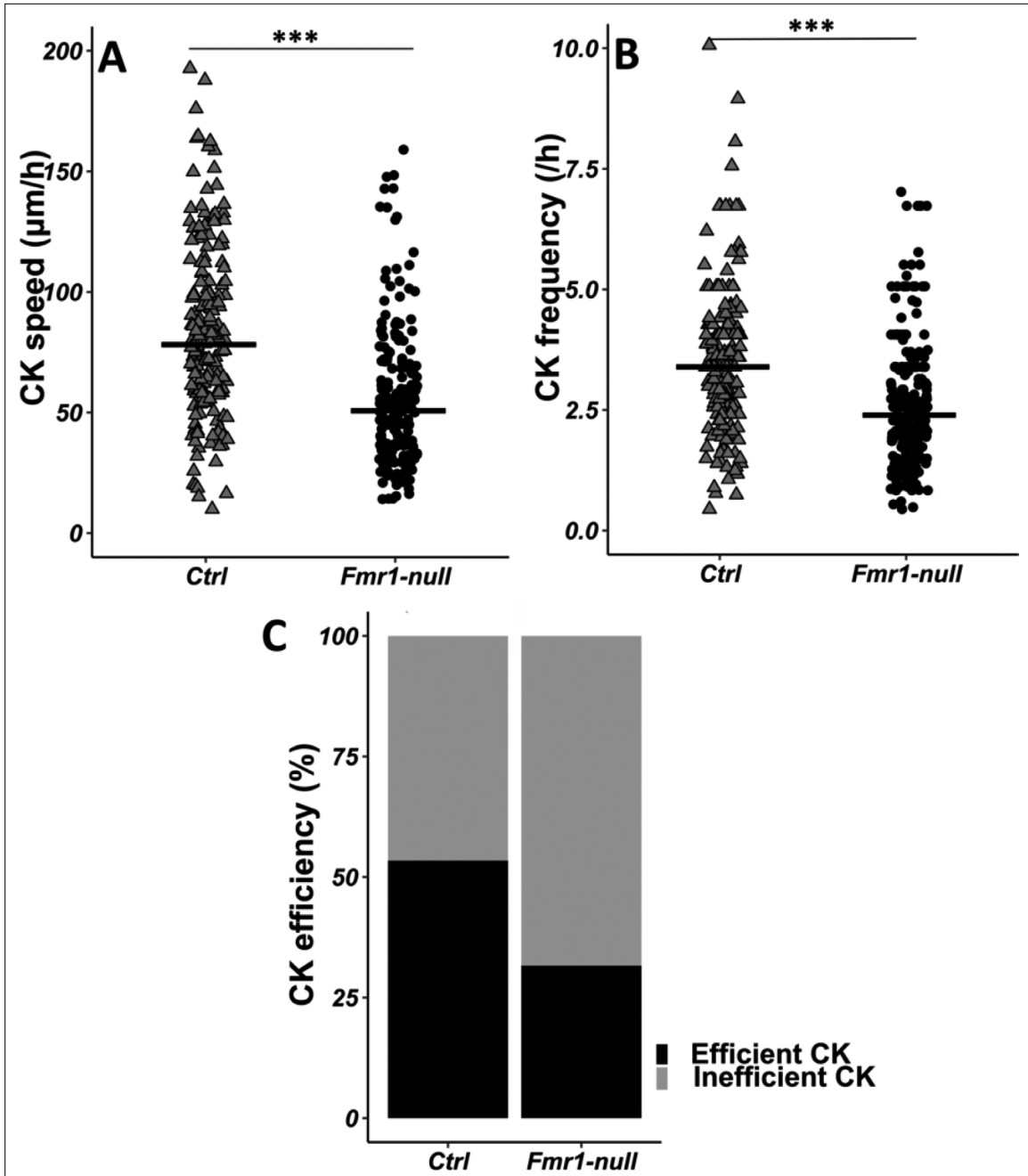

**Figure 3.** Centrokinesis (CK) defects in *Fmr1*-null neurons. (**A**) CK speed of control and *Fmr1*-null neurons. Ctrl: 76.90 (46.46) μm/h; *Fmr1*-null: 49.45 (33.66) μm/hr (Mann-Whitney test, p-value <0.001). (**B**) CK frequency of control and *Fmr1*-null neurons. Ctrl: 3.33 (1.71) CK/hr; *Fmr1*-null: 2.33 (1.71) CK/hr (Mann-Whitney test, p-value <0.001). (**C**) Percentage of efficient CKs in control and *Fmr1*-null neurons. Ctrl: 54%; *Fmr1*-null: 33% (Chi2=57.611, p-value <0.001). The black line represents the median. Ctrl: N=3, n=178; *Fmr1*-null: N=3, n=216. Median (IQR). ***p-value <0.001. Source videos, tracking data (CK speed and efficiency and frequency), and statistical analysis of migrating control and Fmr1-null neurons are available at https://osf.io/eqhzx.

The online version of this article includes the following video for figure 3:

**Figure 3—video 1.** Time-lapse imaging of GFP and centrin-RFP co-electroporated neurons in Fmr1-null rostral migratory stream (RMS).
https://elifesciences.org/articles/88782/figures#fig3video1

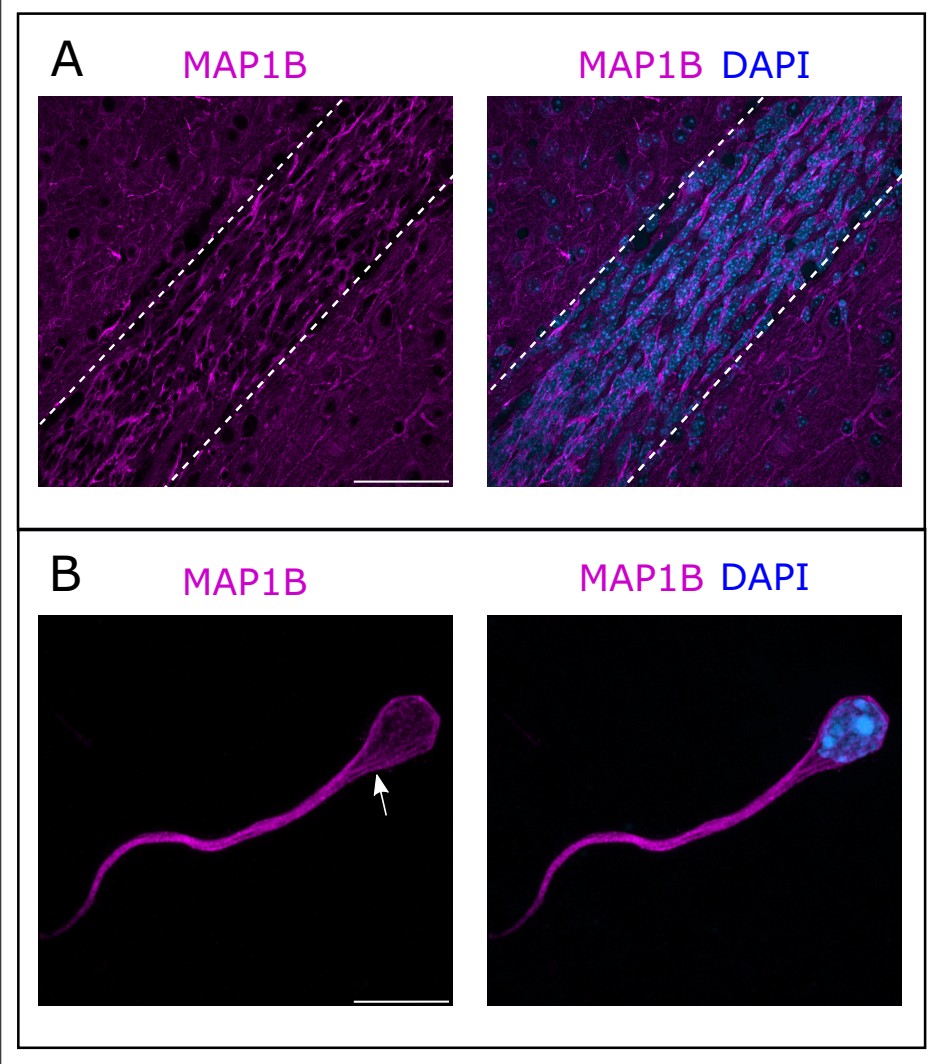

**Figure 4.** Microtubule-associated protein 1B (MAP1B) expression in the rostral migratory stream (RMS).
(**A**) Immunohistochemistry of the RMS showing MAP1B expression (magenta) along the stream. The RMS is delineated with dotted lines. Scale bar: 50 µm. (**B**) Immunostaining for MAP1B of a neuroblast migrating away from a ventricular/subventricular zone (V/SVZ) explant in Matrigel. The labeling is located around the nucleus and in the leading process with occasional bundles of potential microtubules (arrow). Scale bar: 10 µm.

The online version of this article includes the following figure supplement(s) for figure 4:

**Figure supplement 1.** Microtubule-associated protein 1B (MAP1B) is overexpressed in *Fmr1*-null rostral migratory stream (RMS) neurons.

## Discussion

FMRP is commonly described as a pivotal regulator of neuronal plasticity and neural development (*Richter and Zhao, 2021*). However, its role in neuronal migration remains poorly understood.

A role for FMRP in radial embryonic migration was first evidenced by *La Fata et al., 2014*. This study revealed a misplacement of neurons in the embryonic cortical plate in *Fmr1*-null mutants, attributable to disruptions in the multipolar to bipolar transition, a step preceding the main movement of neurons. N-Cadherin emerged as the main FMRP target in this context. It is noteworthy that, while the study did not strongly indicate an impact on radial locomotion per se, drawing conclusive results is challenging due to the relatively low number of analyzed neurons. Subsequently, another study, focusing on miR-129, showed that *Fmr1* KD similarly triggered a misplacement of neurons in the cortical plate. However, this study did not display any further detailed analysis of the precise impact of *Fmr1* KD on radial migration (*Wu et al., 2019*). Furthermore, to our knowledge, there has been

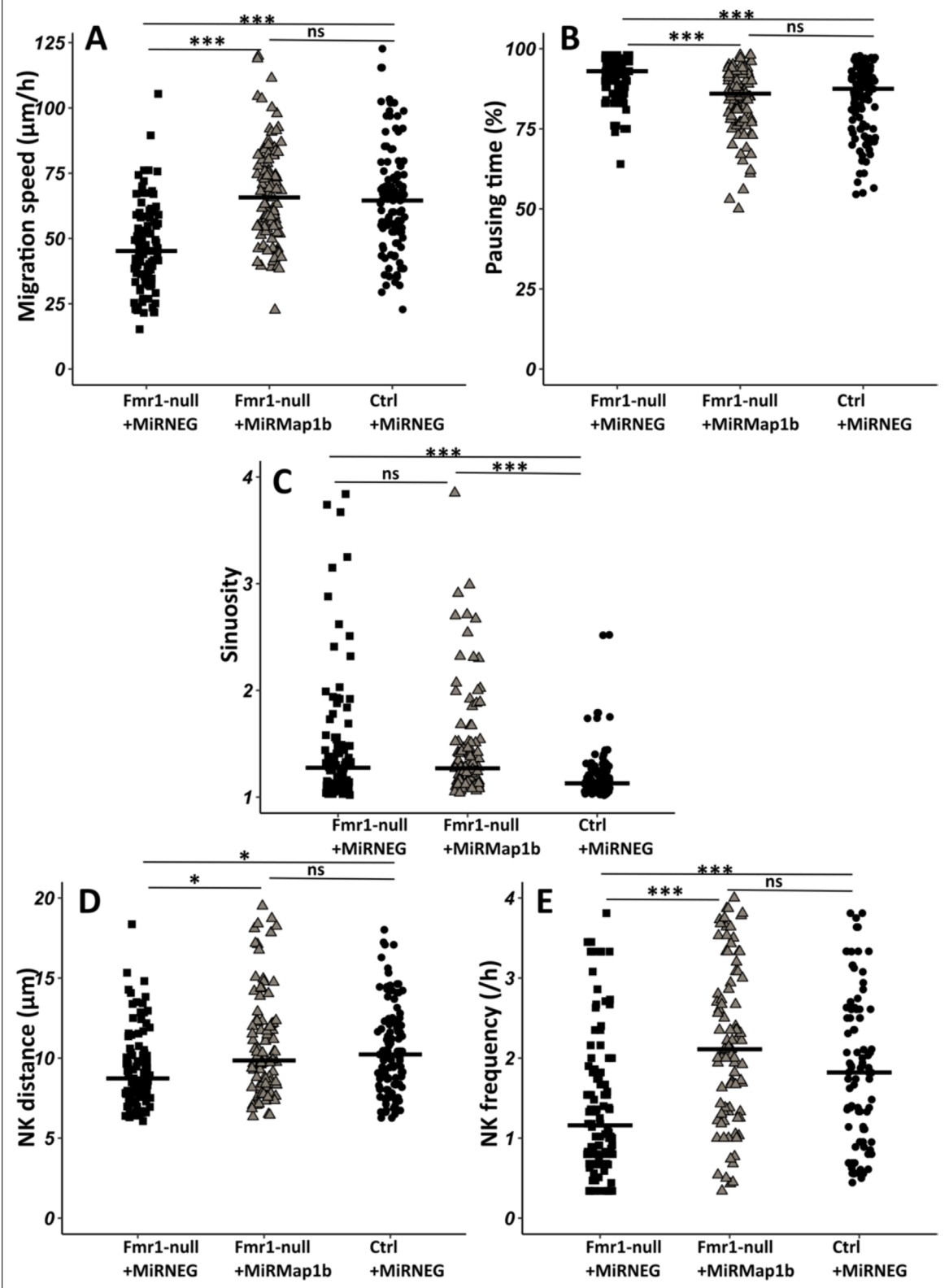

**Figure 5.** Map1b knockdown (KD) rescues *Fmr1*-null neurons migration defects. (**A**) Migration speed of *Fmr1*-null neurons expressing miRNEG and miRMap1b and control neurons expressing miRNEG. *Fmr1*-null neurons + miRNEG: 45.23 (23.25) μm/hr; *Fmr1*-null neurons + miRMap1b: 65.72 (24.31) μm/hr; control neurons + miRNEG: 64.54 (21.99) μm/hr (Kruskall-Wallis test: Chi2=61.168, p-value <0.001, df = 2; followed by Dunn's post hoc test). (**B**) Percentage of pausing time of *Fmr1*-null neurons expressing miRNEG and miRMap1b and control neurons expressing miRNEG. *Fmr1*-null neurons

*Figure 5 continued on next page*

*Figure 5 continued*

+ miRNEG: 93 (7); *Fmr1*-null neurons + miRMap1b: 86 (14); control neurons + miRNEG: 87.50 (15.67) (Kruskall-Wallis test: Chi2=45.716, p-value <0.001, df = 2; followed by Dunn's post hoc test). (**C**) Sinuosity index of *Fmr1*-null neurons expressing miRNEG and miRMap1b and control neurons expressing miRNEG. *Fmr1*-null neurons + miRNEG: 1.30 (0.45); *Fmr1*-null neurons + miRMap1b: 1.28 (0.55); control neurons + miRNEG: 1.13 (0.16) (Kruskall-Wallis test: Chi2=39.807, p-value <0.001, df = 2; followed by Dunn's post hoc test). (**D**) Nucleokinesis (NK) mean distance of *Fmr1*-null neurons expressing miRNEG and miRMap1b and control neurons expressing miRNEG. *Fmr1*-null neurons + miRNEG: 8.93 (3.64) μm; *Fmr1*-null neurons + miRMap1b: 9.89 (3.85) μm; control neurons + miRNEG: 10.23 (3.9) μm (Kruskall-Wallis test: Chi2=11.573, p-value = 0.003, df = 2; followed by Dunn's post hoc test). (**E**) NK frequency of *Fmr1*-null neurons expressing miRNEG and miRMap1b and control neurons expressing miRNEG. *Fmr1*-null neurons + miRNEG: 1.18 (1.11) NK/hr; *Fmr1*-null neurons + miRMap1b:: 2.22 (1.95) NK/hr; control neurons + miRNEG1.65(2) NK/hr (Kruskall-Wallis test: Chi2=39.272, p-value <0.001, df = 2; followed by Dunn's post hoc test). The black line represents the median. *Fmr1*-null neurons + miRNEG: N=6, n=102; *Fmr1*- null neurons + miRMap1b: N=3, n=101; control neurons + miRNEG: N=3, n=78. Median (IQR). *p-value <0.05; ***p-value <0.001; n.s. Source videos, tracking data (NK frequency, pausing time, sinuosity, speed, NK distance), and statistical analysis of migrating contro, Fmr1-null neurons and FMR1-null neurons with miRMAP1B are available at https://osf.io/eqhzx.

The online version of this article includes the following video for figure 5:

**Figure 5—video 1.** Time-lapse imaging of miRMAP1B-GFP electroporated neurons in Fmr1-null rostral migratory stream (RMS). https://elifesciences.org/articles/88782/figures#fig5video1

no exploration of tangential migration in the absence of FMRP, and the impact of its absence on the dynamics of saltatory migration remains unexplored.

We report here significant defects in postnatal tangential migration, characterized by a sloweddown and erratic trajectory. Notably, despite the mutated neurons experiencing delays and challenges in orienting themselves, they eventually reach the OB correctly, as we previously demonstrated in the adult (*Scotto-Lomassese et al., 2011*). Importantly, while a delay in migration may not necessarily trigger important anatomical anomalies, when abnormally migrating neurons ultimately properly reach their target, it is noteworthy that a delay in the timing of differentiation of delayed neurons can lead to significant functional consequences (*Bocchi et al., 2017*).

Live imaging enabled us to conduct a detailed analysis of both NK and centrosome dynamics, revealing profound perturbations in both processes. Given the crucial role of microtubules in regulating these processes (*Kuijpers and Hoogenraad, 2011*; *Tsai and Gleeson, 2005*), our attention turned to MAP1B as the potential key *Fmrp* mRNA target responsible for the observed migratory phenotypes. MAP1B is among the well-established targets of FMRP (*Brown et al., 2001*; *Darnell et al., 2001*; *Zhang et al., 2001*). As a neuron-specific microtubule-associated protein, MAP1B is expressed early in the embryonic brain (*Tucker et al., 1989*) and plays a crucial role in various stages of neural development (*Gonzalez-Billault et al., 2004*) including migration. MAP1B-deficient mice display migration anomalies in the cortex, hippocampus, and cerebellum and MAP1B phosphorylation can be induced by Reelin, a key regulator of radial migration (*González-Billault et al., 2005*).

We show MAP1B overexpression in *Fmr1*-mutated neurons and rescue the migratory defects through RNA-interference-induced KD. This establishes MAP1B as the critical *Fmrp* mRNA target involved in regulating cyclic saltatory migration. The significance of FMRP-regulated MAP1B translation was initially evidenced in *Drosophila*, where it appeared essential for proper synaptogenesis (*Zhang et al., 2001*). This finding was subsequently confirmed in the mouse hippocampus, where elevated MAP1B levels in the *Fmr1*-null context also led to defective synaptogenesis (*Lu et al., 2004*). To our knowledge, the importance of the FMRP-MAP1B duo for neuronal migration, as described here, stands as the only other documented neurodevelopmental function of FMRP-regulated MAP1B translation. Considering the previously reported role of MAP1B in radial migration (*González-Billault et al., 2005*), it is tempting to hypothesize that our findings may extend to this form of embryonic migration, with the duo also regulating its microtubular dynamic.

Remarkably, we also elucidate the mechanistic action of the FMRP/MAP1B duo on the microtubular cytoskeleton. In the absence of FMRP, the cage formed by microtubules around the nucleus of migrating neurons appears disorganized. This disorganization primarily results from MAP1B overexpression in the Fmr1 mutants, as evidenced by the rescue of the phenotype through MAP1B KD. Notably, this phenotype of cage disruption was already described in radially migrating neurons upon Lis1, Dynein, and Ndel1 KD (*Shu et al., 2004*). The Dynein/Lis1/Nde1/Nedl1 complex is essential to neuronal migration, allowing proper traction of the centrosome and nucleus along microtubules (*Tsai et al., 2007*). Considering that MAP1B has also been reported to impact the interaction between

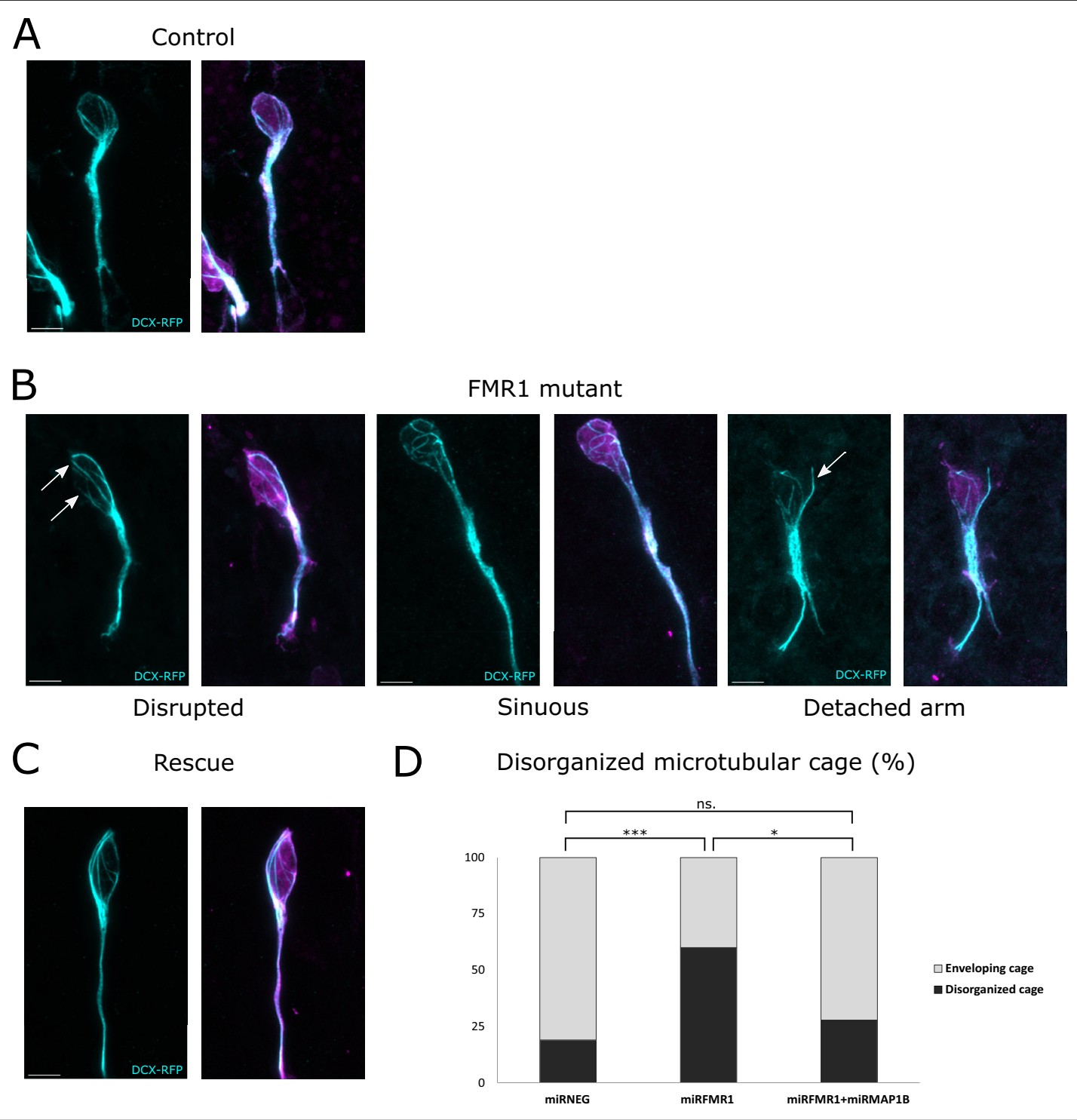

**Figure 6.** The fragile X messenger ribonucleoprotein (FMRP)/microtubule-associated protein 1B (MAP1B) duo acts on the microtubular nuclear cage of rostral migratory stream (RMS) neurons. (**A**) Representative control neurons from mirNEG-GFP (magenta) plus DCX-RFP (cyan) co-electroporated ventricular/subventricular zone (V/SVZ) explants cultured in Matrigel displaying microtubular bundles enveloping the nucleus. (**B**) Representative Fmr1 knockdown (KD) neurons from mirFmr1-GFP (magenta) plus DCX-RFP (cyan) co-electroporated V/SVZ explants cultured in Matrigel. The majority of them display an abnormal cage with disruption (arrows) (46%), sinuous cages (33%), or even detached bundle (arrow) (28%). (**C**) Representative rescued neurons from mirNEG-GFP (magenta) plus DCX-RFP (cyan) plus mirMAP1B co-electroporated V/SVZ explants cultured in Matrigel displaying microtubular bundles enveloping the nucleus. Scale bars: 5 µm. (**D**) Quantification of the percentage of disorganized cages in the different conditions: mirNEG 19%, mirFMR1 60%, mirFMR1 + mirMAP1B 28% (Pearson's $\chi^2$ test [2, N=107])=14.16, p-value <0.001 with the Benjamini-Hochberg method for

*Figure 6 continued on next page*

*Figure 6 continued*

correcting multiple testing. mirNEG, N=3, n=37; mirFMR1 N=3, n=35; mirF+ mirMAP1B, n=36. Acquisitions of Immunostaining of microtubular cage (DCX-RFP) in controls, miRFMR1 and miRFMR1+miRMAP1B, and quantifications of disorganized microtubular cage and statistical analysis are available at https://osf.io/eqhzx.

Dynein and Lis1 (*Jiménez-Mateos et al., 2005*), it is tempting to speculate that MAP1B overexpression in *Fmr1* mutants destabilizes the Dynein/Lis1/Nde1/Nedl1 complex, influencing the link it creates between the nuclear envelope and microtubules (*Gonçalves et al., 2020*) necessary for proper forward traction.

In the context of FXS, it is noteworthy that recent studies have revealed that postnatal tangential migration in the infant human brain is even more extensive than in mice (*Paredes et al., 2016*; *Sanai et al., 2011*). Analyzing migration in FXS human organoids or forthcoming assembloids (*Levy and Paşca, 2023*) will enable to test the conservation of the migratory phenotype observed as well as its molecular underpinnings.

In conclusion, our findings unveil a novel facet of FMRP, highlighting its role as a regulator of migration intricately linked to microtubules through MAP1B. This discovery expands our understanding of this multifaceted protein and sheds a new light on its functional repertoire.

## Materials and methods
### Mouse lines
Mice were housed in a 12 hr light/dark cycle, in cages containing one or two females and one male. The postnatal mice were housed in their parents' cage. Animal care was conducted in accordance with standard ethical guidelines (National Institutes of Health [NIH] publication no. 85-23, revised 1985 and European Committee Guidelines on the Care and Use of Laboratory Animals 86/609/EEC). The experiments were approved by the local ethics committee (Comité d'Ethique en Expérimentation Animale Charles Darwin C2EA-05 and the French Ministère de l'Education Nationale, de l'Enseignement Supérieur et de la Recherche APAFIS#13624-2018021915046521_v5). We strictly performed this approved procedure. The mice used were in a C57BL6-J background. *Fmr1*-null mice were genotyped according to the original protocol (*Bakker et al., 1994*).

### miRNA production
Silencing of *Fmr1* and *Map1b* has been performed using BLOCK-iT Pol II miR RNAi Expression Vector Kits (Invitrogen) and the RNAi Designer (Invitrogen). The sequences of the single-stranded oligos are:

> Fmr1 Top:
> TGCTGTACAAATGCCTTGTAGAAAGCGTTTTGGCCAACTGACTGACGCTTTCTAAGGCATTTGTA,
> Fmr1 Bottom:
> CCTGTACAAATGCCTTAGAAAGCGTCAGTCAGTGGCCAAAACGCTTTCTACAAGGCATTTGTAC,
> Map1b Top:
> TGCTGTGTTGATGAAGTCTTGGAGATGTTTTGGCCACTGACTGACATCTCCAACTTCATCAACA,
> Map1b Bottom:
> CCTGTGTTGATGAAGTTGGAGATGTCAGTCAGTGGCCAAAACATCTCCAAGACTTCATCAACAC.

The double-stranded oligos were inserted in a pcDNA 6.2-GW/EmGFP-miR. The resulting constructions were sequenced before use.

### Plasmids
The plasmids used for that study, in addition to the mir plasmids described above (Invitrogen BLOCK-iT kit), were pCAGGS-GFP (gift from S Garel) and pCMV-centrinRFP (Addgene #26753).

## Postnatal electroporation

Postnatal injection and electroporation were performed at postnatal day 2 (P2). Postnatal mice were anesthetized by hypothermia. Pseudo-stereotaxic injection (from lambda medial-lateral [M/L]: 0.9; anterior-posterior [A/P]: 1.1; dorsal-ventral [D/V]: 2) using a glass micropipette (Drummond Scientific Company, Wiretrol I, 5-000-1050) was performed, and 2 µl of plasmid (between 5 and 8 µg/ml) was injected. Animals were subjected to 5 pulses of 99.9 V during 50 ms separated by 950 ms using the CUY21 SC Electroporator and 10 mm tweezer electrodes (Harvard Apparatus, Tweezertrode, 10 mm, 45-0119). The animals were placed on 37°C plates to restore their body temperature before returning in their parents' cage. Animals were considered as fully recovered when moving naturally and their skin color had returned to pink.

## Postnatal acute brain slices

Brain slices of mice aged from P6 to P10 were prepared as previously described in *Stoufflet et al., 2020*. Pups were sacrificed by decapitation and the brain was removed from the skull. Sagittal brain sections (250 µm) were cut with a VT1200S vibratome (Leica). Slices were prepared in an ice-cold cutting solution of the following composition: 125 mM NaCl, 0.4 mM $CaCl_2$, 1 mM $MgCl_2$, 1.25 mM $NaH_2PO_4$, 26 mM $NaHCO_3$, 5 mM sodium pyruvate, 20 mM glucose, and 1 mM kynurenic acid, saturated with 5% $CO_2$ and 95% $O_2$. Slices were incubated in this solution for 30 min at room temperature and then placed in recording solution (identical to the cutting solution, except that the $CaCl_2$ concentration is 2 mM and kynurenic acid is absent) for at least 30 min at 32°C before image acquisition.

## Time-lapse video microscopy of postnatal slices

To analyze neuronal migration and centrosome dynamics, images were obtained with an inverted SP5D confocal microscope (Leica) using a 40×/1.25-numerical aperture (N.A.) objective with 1.5 optical zoom or an upright SP5 MPII two-photon microscope (Leica) using a 25×/0.95 N.A. objective with 1.86 optical zoom. Images were acquired every 3 min for 2–3 hr. The temperature in the microscope chamber was maintained at 32°C during imaging and brain slices were continuously perfused with heated recording solution (see above) saturated with 5% $CO_2$ and 95% $O_2$.

## Analyses of neuronal migration and centrosome movement

Analyses were performed using ImageJ (NIH Image; National Institutes of Health, Bethesda, MD, USA) software and MTrackJ plugin. The nucleus and the centrosome (when centrin-RFP was co-injected) were tracked manually on each time frame during the whole movie. We considered a NK as a movement superior to 6 µm between two consecutive time points (3 min interval). For cell migration, calculation of speed, percentage of pausing time, sinuosity, directionality, NK distance, and frequency was performed using the x,y,t coordinates of the nucleus of each cell. Cells tracked for less than 30 min and cells that did not perform any NK during the whole tracking were excluded. A CK was defined as a forward movement superior to 2 µm followed by a backward movement superior to 2 µm. For centrosome movement, calculation of CK speed, frequency, and efficiency was performed using the x,y,t coordinates of the centrosome of each cell and the x,y,t coordinates of each corresponding nucleus.

More specifically, *speed* was calculated by summing all the distances traveled by one cell and dividing the total distance by the total time, including time slots (3 min interval) when the cell pauses.

The *pausing time* is calculated as the sum of the time slots during which the cell moves less than 6 µm, hence below the cutoff for an NK.

The *sinuosity index* is the ratio between the total distance covered by the cell and the Euclidean distance between its starting and ending points.

The assessment of *directionality* involves calculating a migration angle (θ), defined as the angle between a neuron's vector from its starting point to its destination and a hypothetical vector connecting the SVZ to the OB. Migration angles range from 0° to 360°, with 0° representing precise alignment with the SVZ-to-OB direction. To simplify, we employ 'migration radars' to depict cell migration angles, segmented for visualizing the distribution of directions.

An *efficient CK* was defined as a forward movement of the centrosome greater than 2 µm followed by a backward movement greater than 2 µm, associated to a subsequent NK (movement of the

nucleus greater than 6 µm). An inefficient CK was defined by the same centrosomal movement not followed by an NK.

## Immunohistochemistry

P7 to P10 mice were lethally anesthetized using Euthasol. Intracardiac perfusion with 4% paraformaldehyde was performed. The brain was post-fixed overnight in 4% paraformaldehyde and then rinsed three times with phosphate-buffered saline (PBS) 1× (Gibco, 1400-067). 50 µm sagittal sections were made with VT1200S microtome (Leica). Slices were placed 1 hr in a saturation solution (10% fetal bovine serum; 0.5% Triton X-100 in PBS). Primary antibodies used in this study are: GFP (Aves; GFP-1020; 1/1000), FMRP (Developmental Studies Hybridoma Bank; 2F5-1; 1/200), MAP1B (Santa Cruz Biotechnology; sc-365668; 1/300). The antibodies were diluted in saturation solution. Slices were incubated for 48–72 hr at 4°C under agitation with the antibodies and then rinsed three times with PBS 1×. Secondary antibodies used are: anti-chicken immunoglobulin Y (IgY) Alexa Fluor 488 (1/1000; Jackson ImmunoResearch; 703-545-155) against anti-GFP, anti-mouse immunoglobulin G2b (IgG2b) Alexa Fluor 555 (1/2000; Jackson ImmunoResearch; 703-545-155) against anti-FMRP and anti-MAP1B. The antibodies were diluted in saturation solution. Slices were incubated with secondary antibodies for 1 hr at room temperature under agitation, protected from light. After rinsing three times with PBS 1×, slices were counter-colored with Hoechst and mounted in Mowiol.

## Immunostaining on SVZ explants in Matrigel

For the quantification of mirFmr1 and miR*Map1b* efficiency, the SVZ of electroporated mice were dissected as described. SVZ explants were placed on glass-bottom culture dishes (MatTek Corporation; P35G-0-20C) within 10 ml of 60% Matrigel (Corning; 356237). After Matrigel solidification (15 min at 37°C, 5% $CO_2$), culture medium was added and the dishes were incubated for 4–5 days at 37°C, 5% $CO_2$. For FMRP and MAP1B immunostaining, SVZ cultures were fixed in 2% paraformaldehyde for 30 min and then rinsed three times with PBS 1×. Immunostaining was then performed as for sections (see above). To quantify *Fmr1* and *Map1b* KD, a cell was considered MAP1B or FMRP negative when it was clearly immunonegative at high magnification.

## Tissue collection and western blotting

RMS were manually micro-dissected from five to six postnatal mouse brains and pooled in a PBS 1× (0.5% glucose) solution. After centrifugation, protein extraction was performed on the tissue pellet. Samples were homogenized in a lysis buffer 25 mM Tris HCl pH 7.5, 150 mM NaCl, 1% NP40, 0.5% Na deoxycholate, 1 mM EDTA, 5% glycerol, 1× EDTA-free protease inhibitor cocktail (Sigma, 4693132001). After centrifugation, samples were loaded and run on NuPAGE 3–8% Tris-Acetate Gel (Invitrogen, EA0378BOX) at 120 V for 15 min then 180 V for 40 min. Transfer to nitrocellulose Immobilon-PVDF-P membrane (Millipore, IPVH00010) was performed at 40 V overnight at 4°C. The membrane was then saturated for 1 hr in TBST containing 5% powder milk. Primary antibodies used are: MAP1B (Santa Cruz Biotechnology, sc-365668, 1/100), Vinculin (Cell Signaling Technology, 13901 S, 1/1000). The antibodies were diluted in TBST containing 5% powder milk. Secondary antibodies used are: ECL anti-mouse immunoglobulin G (IgG) horseradish peroxidase linked whole antibody (1/10,000; Fisher Scientific; NXA931V) for anti-MAP1B, Peroxidase-conjugated AffiniPure F(ab')2 Fragment Donkey Anti-Rabbit IgG (H+L) (1/5000; Jackson ImmunoResearch; 711-036-152) for anti-Vinculin. The antibodies were diluted in TBST containing 5% powder milk. Labeling was visualized using Pierce ECL Western Blotting Substrate (Thermo Scientific; 32209) and luminescent image analyzer LAS-3000.

## Statistics

All manipulations and statistical analyses were implemented with R (4.2.1, R Foundation for Statistical Computing, Vienna, Austria). Normality in the variable distributions was assessed by the Shapiro-Wilk test. Furthermore, the Levene test was performed to probe homogeneity of variances across groups. Variables that failed the Shapiro-Wilk or the Levene test were analyzed with non-parametric statistics using the one-way Kruskal-Wallis analysis of variance on ranks followed by Dunn's post hoc test and Mann-Whitney rank sum tests for pairwise multiple comparisons. Variables that passed the normality test were analyzed by means of one-way ANOVA followed by Tukey's post hoc test for multiple comparisons or by Student's t test for comparing two groups. Categorical variables were

compared using Pearson's Chi2 test or Fisher's exact test. A p-value of <0.05 was used as a cutoff for statistical significance. Results are presented as the median (interquartile range [IQR]). The statistical tests are described in each figure legend.

## Acknowledgements

We thank Isabelle Dusart for expert reading as well as Caroline Dubacq and Oriane Trouillard for technical help. The experiments were performed in IBPS imaging facility and the mice were housed in IBPS animal facility. This work was funded by Fondation Jérôme Lejeune, ANR NotifX ANR-20-CE16-0016 and NIH contract NIDCD grant R01-DC-017989.

## Additional information

### Funding

| Funder | Grant reference number | Author |
|---|---|---|
| Fondation Jérôme Lejeune | GRT-2017A/1659 | Isabelle Caille |
| Agence Nationale de la Recherche | ANR-20-CE16-0016 | Isabelle Caille |
| National Institute on Deafness and Other Communication Disorders | R01-DC-017989 | Alain Trembleau |

The funders had no role in study design, data collection and interpretation, or the decision to submit the work for publication.

### Author contributions

Salima Messaoudi, Conceptualization, Formal analysis, Investigation, Methodology, Writing - original draft; Ada Allam, Theo Paillard, Conceptualization, Investigation, Methodology, Writing - review and editing; Julie Stoufflet, Conceptualization, Formal analysis, Investigation; Anaïs Le Ven, Investigation; Coralie Fouquet, Mohamed Doulazmi, Investigation, Methodology; Alain Trembleau, Funding acquisition, Validation; Isabelle Caille, Conceptualization, Formal analysis, Supervision, Funding acquisition, Investigation, Methodology, Writing - original draft, Writing - review and editing

### Author ORCIDs

Ada Allam http://orcid.org/0009-0008-1171-5323
Alain Trembleau http://orcid.org/0000-0002-8290-0795
Isabelle Caille http://orcid.org/0000-0002-9054-2641

### Ethics

Animal care was conducted in accordance with standard ethical guidelines [National Institutes of Health (NIH) publication no. 85-23, revised 1985 and European Committee Guidelines on the Care and Use of Laboratory Animals 86/609/EEC]. The experiments were approved by the local ethics committee (Comité d'Ethique en Expérimentation Animale Charles Darwin C2EA-05 and the French Ministère de l'Education Nationale, de l'Enseignement Supérieur et de la Recherche APAF-IS#13624-2018021915046521_v5). We strictly performed this approved procedure.

Reviewer #1 (Public Review): https://doi.org/10.7554/eLife.88782.3.sa1
Reviewer #3 (Public Review): https://doi.org/10.7554/eLife.88782.3.sa2
Author response https://doi.org/10.7554/eLife.88782.3.sa3

## Additional files

### Supplementary files
• MDAR checklist

## Data availability

All data generated or analysed during this study are included in the manuscript. Source data files for Figures 2, 3, 5, 6, and Figure 2—figure supplements 1, 2 and Figure 4—figure supplement 1 are available at the following link: https://osf.io/eqhzx/.

The following dataset was generated:

| Author(s) | Year | Dataset title | Dataset URL | Database and Identifier |
|---|---|---|---|---|
| Allam A, Caillé I | 2024 | FMRP regulates postnatal neuronal migration via MAP1B | https://osf.io/eqhzx | Open Science Framework, eqhzx |

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
