## [Editor Report · eLife assessment]

This study addresses the role of FMRP in the migration of newborn neuroblasts in the postnatal brain. Through extensive and **convincing** analysis of living imaging videos, the authors showed that neurons with FMRP deletion migrate aberrantly and exhibit defects in nucleokinesis and centrokinesis. The study presents a **valuable** finding on the mechanism of neuroblast migration in the postnatal brain.

---

## [Referee Report · Reviewer #1 (Public Review)]

This study investigated Fragile X Messenger Ribonucleoprotein (FMRP) protein impact on neuroblast tangential migration in the postnatal rostral migratory stream (RMS). Authors conducted a series of live-imaging on organotypic brain slices from Fmr1-null mice. They continued their analysis silencing Fmr1 exclusively from migrating neuroblasts using electroporation-mediated RNA interference method (MiRFmr1 KD). These impressive approaches show that neuroblasts tangential migration is impaired in Fmr1-null mice RMS and these defects are mostly recapitulated in the MiRFmr1neuroblasts. This nicely supports the idea that FMRP have a cell autonomous function in tangentially migrating neuroblasts. Authors also confirm that FMRP mRNA target Microtubule Associated Protein 1B (MAP1B) is overexpressed in the Fmr1-null mice RMS. They successfully use electroporation-mediated RNA interference method to silence Map1b in the Fmr1-null mice neuroblasts. This discreet and elaborate experiment rescues most of the migratory defects observed both in Fmr1-null and MiRFmr1neuroblasts. Altogether, these results strongly suggest that FMRP-MAP1B axis has an important role in regulation of the neuroblasts tangential migration in the RMS. Neurons move forward in cyclic saltatory manner which includes repeated steps of leading process extension, migration of the cell organelles and nuclear translocation. Authors reveal by analyzing the live-imaging data that FMRP-MAP1B axis is affecting movement of centrosome and nucleus during saltatory migration. An important part of the centrosome and nucleus movement is forces mediated by microtubule dynamics. Authors propose that FMRP regulate tangential migration via microtubule dynamics regulator MAP1B. This work provides valuable new information on regulation of the neuroblasts tangential saltatory migration. These findings also increase and improve our understanding of the issues involved in Fragile X Syndrome (FXS) disorders. The conclusions of this work are supported by the presented data.

The current version of the study has improved substantially. Authors have enhanced the material and methods section including a more detailed section on the neuronal migration analysis. This amendment is a very valuable addition and strengthens the interpretation of the results, analysis and conclusions. Authors also have strengthened and clarified their results providing a more profound analysis of the migration directionality between controls, Fmr1-null, MiRFmr1 KD and MiRMap1b KD neuroblasts. They have incorporated new results in the study which elaborate FMRP and MAP1B participation in microtubule organization during tangential migration. Authors show that FMRP-MAP1B axis act on microtubule cage surrounding the nucleus. Microtubule cage participate on proper nuclear movement during neuron migration. These results emphasize more the interplay between FMRP, MAP1B, and the microtubule cytoskeleton. The authors have successfully expanded both the introduction and discussion sections of the manuscript.

---

## [Referee Report · Reviewer #3 (Public Review)]

Neuronal migration is one of the key processes for appropriate neuronal development. Defects in neuronal migration are associated with different brain disorders often accompanied by intellectual disabilities. Therefore, the study of the mechanisms involved in neuronal migration helps to understand the pathogenesis of some brain malformations and psychiatric disorders.

FMRP is an RNA-binding protein implicated in RNA metabolism regulation and mRNA local translation. FMRP loss of function causes fragile X syndrome (FXS), the most common form of inherited intellectual disability. Previous studies have shown the role of FMRP in the multipolar to bipolar transition during the radial migration in the cortex and its possible relation with periventricular heterotopia and altered synaptic communication in humans with FXS. However, the role of FMRP in neuronal tangential migration is largely unknown. In this manuscript, the authors aim to decipher the role of FMRP in the tangential migration of neuroblasts along the rostral migratory stream (RMS) in the postnatal brain. By extensive live-imaging analysis of migrating neuroblasts along the RMS, they demonstrate the requirement of FMRP for neuroblast migration and centrosomal movement. These migratory defects are cell-autonomous and mediated by the microtubule-associated protein Map1b.

Overall, the manuscript highlights the importance of FMRP in neuronal tangential migration. They performed an analysis of different aspects of migration such as nucleokinesis and cytokinesis in migrating neuroblasts from live-imaging videos. The authors have reinforced the results that associate defects in microtubule organization in Fmrp1 KO neurons and this rescue with the microtubule-associated protein Map1b. Overall, results concerning the role of Fmr1 in the tangential migration of neuroblasts are solid and convincing.

However, the work is still quite incomplete. My main concern is still what are the functional consequences of delay in neuroblast migration in the integration and function of OB interneurons and this relation with FXS pathophysiology. An anatomical examination of the RMS in the Fmr1KO mice is still missing.

---

## [Author Response]

The following is the authors’ response to the original reviews.

**To Reviewer #1**

We sincerely appreciate the constructive and insightful comments provided by the reviewer. Their valuable suggestions have been meticulously considered, leading to comprehensive modifications within the article.

In addition, we want to stress that we have implemented a significant additional modification by introducing a new figure (Fig. 6). This figure highlights the collaborative impact of FMRP and Map1B on the microtubular structure of migrating neurons. We firmly believe that this molecular elucidation of the migration phenotype constitutes a noteworthy addition to our work.

**Public Review**

(1) We have taken the necessary steps to enhance the material and methods section of our neuronal migration analysis. We apologize for any initial lack of detail, including the omission of information on sinuosity index and directionality radar. Regarding the query about speed, we want to clarify that it indeed encompasses the percentage of pausing time. The speed is calculated by dividing the total distance traveled by the cell by the total time it migrated.

(2) We would like to provide a clarification regarding the statistical analysis in our figures. The figures now represent the median, and the legend indicates the median along with the interquartile range. This approach is in line with the use of non-parametric analysis for variables that do not adhere to a normal distribution. Regrettably, in the previous version, there was an oversight in the figure legends where the mean, along with the standard error of the mean, was incorrectly stated instead of the intended representation of the median. We sincerely apologize for any confusion this may have caused. Moving forward, the corrected legend now accurately reflects the statistical measures used in the analysis.

The global Kruskal Wallis analysis, followed by Dunn’s post hoc analysis, does indeed indicate that Fmr1 KD globally replicates the Fmr1-null phenotype. However, we concur with the reviewer's point regarding directionality, and we apologize for any lack of precision in the initial version. Upon further analysis, we have identified a significant difference in directionality (Fisher test p < 0.001). This more pronounced directionality defect in the KD could potentially be indicative of a lack of compensation, a factor that may not be at play in the Fmr1 null context. We appreciate the opportunity to address this issue and our revised version includes the necessary details to accurately convey these findings.

(3) We appreciate the referee's agreement with our perspective.

(4) In response to the recommendations from all referees, we have expanded both the introduction and discussion sections of our manuscript. The initial brevity of these sections was due to the short format we had initially chosen. We believe that these expansions contribute to a more comprehensive and nuanced presentation of our work, addressing the concerns raised by the referees.

**Recommendations for the authors**

The time stamp and scale bars were added.

The median versus mean issue is addressed above.

Figure numbering has been corrected (sorry for the mistake). The efficiency of CK is defined in the Mat and Met section.

**To Reviewer #2**

**Public review**

We express our gratitude to the referee for their positive appreciation of our work. We have carefully considered their suggestions and have modified the article accordingly.

In addition, as said to Referee #1, we want to stress that we have implemented a significant additional modification by introducing a new figure (Fig. 6). This figure highlights the collaborative impact of FMRP and Map1B on the microtubular structure of migrating neurons. We firmly believe that this molecular elucidation of the migration phenotype constitutes a noteworthy addition to our work.

**Recommendations for the authors**

(1) In light of the referee's recommendation, we conducted more resolutive staining of FMRP in SVZ neurons cultured in Matrigel, providing a more precise depiction of its subcellular localization (see Figure 1). Additionally, we have removed the sentence referring to growth cone staining, as it was not visibly present in cultured neurons. We appreciate the guidance from the referee in refining our study.

(2) We have also added a new figure 4 with better staining of MAP1B in the RMS as well as a more resolutive MAP1B staining in cultured neurons.

With all due respect, we maintain that the western blot experiments, conducted in three independent experiments, unequivocally support the conclusion of a 1.6X increase in MAP1B in the RMS of Fmr1null mutants, a trend observed in other systems.

In accordance with the referee's suggestion, we endeavored to quantify RMS immunostainings. Regrettably, the results proved inconclusive. This outcome is not entirely unexpected, as immunostainings are recognized for their inherent challenges in quantification. The additional complexity introduced by neonate perfusion further contributes to the notable interindividual variability observed.

(3) The efficiency of the two interfering RNAs is now documented in the text.Regarding the directionality radar, as highlighted for Ref 1 (public review, point #2), we acknowledge that, while Fmr1KD generally recapitulates the migratory phenotype of the Fmr1 mutants, more precise statistical analysis reveals differences in directionality, which is now documented. We apologize for the previous lack of precision.

(4) The suggested experiment of overexpression is interesting but we faced challenges in its execution. Attempts to overexpress MAP1B through intraventricular electroporation of a CMV-MAP1B plasmid resulted in the immobilization of transfected cells in the SVZ, hindering further analysis of migration. We hypothesize that this outcome may be attributed to a discrepancy in the actual dosage of MAP1B in the mutants.

(5) Concerning this point, and as mentioned above, we have incorporated a crucial piece of information into the manuscript, presented in Figure 6. The data reveal a severe disruption in the microtubular cage surrounding the nucleus of migrating neurons in Fmr1 mutants, a phenomenon rescued by MAP1B knock-down. Based on these findings, we believe we can confidently conclude that the microtubule-dependent functions of MAP1B play a role in the migratory phenotype of Fmr1 mutants. We consider this experiment to be a highly valuable addition to our work, shedding light on the underlying molecular mechanisms.

**To Reviewer #3**

We thank the referee for their insightful comments and have taken their consideration with great considerations.

In addition and as said above, we want to stress that we have implemented a significant additional modification by introducing a new figure (Fig. 6). This figure highlights the collaborative impact of FMRP and Map1B on the microtubular structure of migrating neurons. We firmly believe that this molecular elucidation of the migration phenotype constitutes a noteworthy addition to our work.

**Public review**

With regard to the perceived 'incompleteness' of our work, we believe that the addition of Figure 6, illustrating the molecular underpinnings of the Fmr1 mutation on the microtubular cytoskeleton and its rescue in the MAP1B KD, significantly enhances the completeness of our study.

In response to the comment on the introduction and discussion sections, we acknowledge that their brevity was due to the Short Format initially chosen. We have since expanded these sections, incorporating additional information about FMRP and MAP1B and their influences on migration.

Regarding the La Fata article, as highlighted in our discussion, it's important to note that while the study did not strongly indicate an impact on radial locomotion per se, drawing conclusive results is challenging due to the relatively low number of analyzed neurons. Consequently, we do not believe that it poses a challenge to our findings.

With respect to MAP1B overexpression, as previously mentioned in response to Ref #2, point 4, our attempts resulted in the inhibition of migration, potentially due to an overdosage of the protein.

In terms of anatomical consequences, as highlighted in our discussion, while our neurons experience a delay in migration, they eventually reach their destination. Although a delay in migration may not directly result in significant anatomical anomalies, we acknowledge that the timing of differentiation can be crucial. As noted by Bocchi et al. (2017), a delay in the timing of differentiation for neurons reaching their target could lead to notable functional consequences. In any case, we have tOned down any references to the implication for the pathology.

**Recommendation for the authors**

The size of the figures has been modifiedThe pausing time and sinuosity are now definedThe centrin-RFP labeling was indeed too weak in the previous version, which we corrected. We apologize for this.Fig S3 has been revised to address concerns. Notably, the decision to present the two bands for Vinculin and MAP1B separately is intentional. The blot is cut to allow independent development due to the substantial difference in their development times. We believe this approach provides a more accurate representation of the data.The numbering of the figures has been corrected. Sorry for the initial mistake.The Mat and Meth section has been corrected. Please note that we did not use any culture insert in this study.The tittle has been modifiedComments about the Map1B overexpression experiment are expressed above and in replies to ref #2.